# Cell Culture Models for the Study of Hepatitis D Virus Entry and Infection

**DOI:** 10.3390/v13081532

**Published:** 2021-08-03

**Authors:** Margaux J. Heuschkel, Thomas F. Baumert, Eloi R. Verrier

**Affiliations:** 1Université de Strasbourg, Inserm, Institut de Recherche sur les Maladies Virales et Hépatiques UMR_S1110, 67000 Strasbourg, France; margaux.heuschkel@etu.unistra.fr (M.J.H.); thomas.baumert@unistra.fr (T.F.B.); 2Institut Hospitalo-Universitaire, Pôle Hépato-Digestif, Nouvel Hôpital Civil, 1 Place de L’Hôpital, 67000 Strasbourg, France

**Keywords:** viral cell entry, hepatitis B virus (HBV), hepatitis D virus (HDV), sodium taurocholate co-transporting polypeptide (NTCP), hepatoma cell lines, primary hepatocytes

## Abstract

Chronic hepatitis D is one of the most severe and aggressive forms of chronic viral hepatitis with a high risk of developing hepatocellular carcinoma (HCC). It results from the co-infection of the liver with the hepatitis B virus (HBV) and its satellite, the hepatitis D virus (HDV). Although current therapies can control HBV infection, no treatment that efficiently eliminates HDV is available and novel therapeutic strategies are needed. Although the HDV cycle is well described, the lack of simple experimental models has restricted the study of host–virus interactions, even if they represent relevant therapeutic targets. In the last few years, the discovery of the sodium taurocholate co-transporting polypeptide (NTCP) as a key cellular entry factor for HBV and HDV has allowed the development of new cell culture models susceptible to HBV and HDV infection. In this review, we summarize the main in vitro model systems used for the study of HDV entry and infection, discuss their benefits and limitations and highlight perspectives for future developments.

## 1. Introduction

Chronic hepatitis D constitutes one of the most severe and aggressive forms of chronic viral hepatitis with an accelerated development of liver disease towards fibrosis, cirrhosis, and a high risk of developing hepatocellular carcinoma (HCC) compared to chronic hepatitis B. It is caused by the hepatitis delta virus (HDV), a small hepatotropic virus discovered in Italy in 1977 [1] belonging to the *Deltavirus* genus [2]. HDV is not only the smallest RNA virus known to interact with a human host, but also an HBV satellite virus whose production and release of viral particles depend on the expression of HBV envelope proteins [3]. Therefore, hepatitis D is the result of either HBV and HDV co-infection or HDV/HBV superinfection in chronically HBV-infected patients [4].

According to a recent meta-analysis, the number of people infected with HDV has been estimated to approximately 12 million worldwide [5]. However, due to significant limitations in reliable epidemiological data and varying geographic distribution, this number is most likely to be underestimated. Other studies have suggested a total of 50 to 72 million cases worldwide with variable prevalence across regions, increasing in developed countries despite the existence of a preventive HBV vaccine that also protects against HDV infection [6,7]. 

The 35-nm HDV virion is composed of a ribonucleoprotein (RNP) core complex and an HBV-derived envelope. The RNP complex contains an approximately 1.7 kb, single-stranded, circular, and negative-sense RNA genome associated with the two isoforms of the delta antigen (HDAg): small (S-HDAg) and large (L-HDAg). This genome is highly paired in a characteristic rodlike structure similar to plant viroid genomes [8,9]. During virus replication, two other forms of HDV RNA are produced: the replication intermediate antigenomic RNA and the two forms of HDV mRNA leading to the production of S- and L-HDAg. Both genomic and antigenomic RNA exhibit a ribozyme activity [10]. The virus envelope is composed of an endoplasmic reticulum (ER)-derived lipid bilayer embedding the three HBV envelope proteins: small (S-), medium (M-), and large (L-) HBsAg. These three proteins share a common C-terminal domain named S, corresponding to the small surface antigen (S-HBAg). The medium antigen (M-HBAg) carries an additional hydrophilic N-terminal domain named PreS2. Finally, the N-terminal part of the large antigen (L-HBAg) is characterized by the presence of an additional PreS1 domain [11].

Despite recent progress in the understanding of HDV infection, the treatment of chronic hepatitis D remains challenging. Indeed, HDV is highly dependent on host factors for the completion of its replication cycle. It encodes only one structural protein excluding any polymerase activity that could be targeted by direct antiviral therapy. In addition, attempts to use inhibitors targeting HDV ribozyme activity have been limited by a marked toxicity observed in vitro [12]. Moreover, HDV replication is insensitive to nucleos(t)ide analogues used against HBV [13]. Pegylated interferon (peg-IFN)-based therapies, which have been commonly used as a treatment for HDV infection for decades, have numerous side effects and fail to inhibit HBsAg expression, thus causing a rebound of HDV viral load in most patients after treatment cessation [14]. Although current treatments for hepatitis D are not completely satisfying, the discovery of sodium-taurocholate co-transporting polypeptide (NTCP) as a bona-fide receptor for HBV and HDV in the last few years has opened new perspectives for a better understanding of the viral cycle, whose different steps, including the virus entry in the cells, may constitute relevant therapeutic targets against HDV infection [15,16]. Indeed, direct host-targeting agents (HTAs) such as the HDV/HBV entry inhibitor Bulvertide (Mycludex B/Hepcludex) which, in July 2020, received conditional marketing authorization in the European Union as the first HDV-specific drug [17,18] or the molecule Lonafarnib, which blocks the interaction between L-HDAg and HBsAg during HDV assembly, represent a suitable and robust option for the development of novel therapeutic strategy [19]. This approach, however, requires a deep understanding of the molecular interactions between the virus and its host, which has been impaired for decades in the case of HDV due to the lack of robust easy-to-use cell culture system for the study of its life cycle. Notably, the molecular mechanisms of virus entry into cells have remained poorly understood until recently.

## 2. Molecular Virology of HDV Entry

HDV entry into hepatocytes is a multistep process. As HDV and HBV share the same envelope proteins, it is likely that both viruses enter the cells through a common pathway [11]. In this context, it has to be noticed that HDV has been widely studied as a surrogate model for the study of viral entry, given its replication characteristics, which make it an easier virus to handle than HBV. Their infectivity depends on two determinants, one of which is specific to the L-HBAg antigen, and more particularly to the seventy-five myristoylated residues of the N-terminal sequence of PreS1 domain. The second determinant is located on the antigenic loop (AGL) of the S-HBAg protein [11,20,21,22,23]. 

HDV entry requires an essential first step of attachment of the viral particle to the hepatocyte surface through a low-affinity interaction to heparan sulfate proteoglycans (HSPGs), including Glypican 5 [24]. The virus binding to HSPGs is mediated by electrostatic interactions between the negatively charged HSPG and two positively charged residues (Arg122 and Lys141) in the AGL region of the S domain present in all HBV envelope proteins [25]. Therefore, this low-affinity interaction stabilizes the virus at the cell surface and promotes its high-affinity binding via the PreS1 domain to the virus receptor. NTCP is encoded by *SLC10A1* gene and is exclusively expressed at the basolateral membrane of hepatocytes, suggesting that HDV and HBV hepatotropism strongly relies on its expression pattern [26]. The regulation of NTCP membrane expression is notably controlled by post-translational mechanisms [27] and regulated by several host factors, such as E-cadherin, a calcium-dependent cell–cell adhesion protein capable of binding to glycosylated NTCP, facilitating its relocalization to the basolateral plasma membrane [28].

Assuming a comparable entry mechanism for HBV and HDV, HDV entry into the cell would then be mediated through clathrine-dependent endocytosis [29] involving the epidermal growth factor receptor (EGFR), acting as a host-entry co-factor triggering HBV internalization [30]. It must be noted that these mechanisms were described for HBV entry and would need to be properly validated using HDV infection models to draw a conclusion. 

Following endocytosis, enveloped viruses usually continue along the endocytic pathway. For HBV and HDV, this mechanism is not yet fully understood. EGFR activation has been shown to trigger HBV transport to late endosomes/lysosomes. Thus, all these data suggest that HBV/HDV are co-transported with EGFR and NTCP to late endosomes. However, the signals triggering endosomal fusion remain unknown [31]. 

Although several elements of the HDV and HBV entry process into hepatocytes remain to be elucidated, such as the presence of other entry co-factors, the recent characterization of the virus receptor has killed two birds with one stone. First, it provided a highly valuable target for antiviral therapy. Second, it allowed the development of novel simple in vitro models for the understanding of the full life cycle. These NTCP-overexpressing cell lines represent an attractive complement to the previously existing models, opening a promising perspective for the near characterization of HDV–hepatocyte interactions and novel antiviral targets. Here, we summarize the current models available for the study of HDV entry and infection and discuss their strengths and limitations.

## 3. Primary Hepatocytes (PHH and PTH)

Primary human hepatocytes (PHHs) are the natural hosts of HBV and HDV and are susceptible and permissive to both virus infections [32]. However, the investigation of HBV/HDV infection in PHHs is restricted by a series of constraints (Table 1). First, their supply is limited. Moreover, PHHs are difficult to manage in culture conditions, do not expand, have a limited lifetime, and rapidly lose their hepatocyte-specific characteristics in culture, such as polarization and the expression of critical hepatocyte markers for HDV infections such as NTCP [33]. This observation may explain why primary hepatocytes are susceptible to HBV and HDV for only a few days after isolation. Furthermore, in addition to restrictive culture conditions including the use of dimethyl sulfoxide (DMSO), the susceptibility of PHHs to virus infection is highly donor-to-donor dependent. Consequently, the number of reproducible studies is limited [32]. Nevertheless, the development of novel technologies in the last few years allowing the long-term culture of primary hepatocytes may help to provide a more stable system for the investigation of viral infection [34]. 

HBV and HDV can also infect primary cultures of *Tupaia belangeri* hepatocytes (PTH) [35,36]. The use of PTH allowed Yan et al. to elegantly identify NTCP as a receptor for HDV, by characterizing the interactome of a synthetic HBV PreS1-derived peptide, known to bind the virus receptor at the cell surface, thus demonstrating their suitability and high interest for the study of virus entry and infection [15]. 

To conclude, despite their limitations, primary cultures of hepatocytes are the most relevant in vitro model for the study of HDV infection. Indeed, they exhibit normal hepatic functions such as hepatocyte polarization and the whole presence of hepatic host factors. In addition, they possess a fully functional innate immune system which allows their use to validate HDV-related host factors and to confirm the activity of antiviral molecules [37]. 

## 4. Differentiated HepaRG Cells 

The HepaRG cell line is an immortalized hepatic progenitor cell line derived from a hepatitis C virus (HCV)-induced liver tumor [38]. This cell line has the particularity of conserving a significant number of liver functions and exhibiting a transcriptomic profile comparable to hepatocytes, including the expression of innate immune system factors [39]. For this reason, these cells are widely used for drug metabolism and toxicology tests [37]. Nevertheless, HepaRG cells require a long-term DMSO-mediated differentiation process to acquire susceptibility to HDV infection. These cells are then able to support viral entry and replication, which make them a suitable model for the study of many steps in the HDV life cycle and new drug screening [40]. Notably, the use of the HDV/HepaRG infection system has greatly enhanced the understanding of HBV and HDV entry into cells before the characterization of the receptor, notably demonstrating the importance of virion attachment to HSPGs in the initiation of HDV entry [41]. Taking advantage of the critical step of cell differentiation process in the susceptibility to HDV infection, Ni et al. independently discovered NTCP as a key receptor for both HBV and HDV [16]. By comparing the transcriptomic pattern of non-differentiated and differentiated cells, they isolated NTCP as a main candidate expressed at the cell membrane, exclusively in hepatocytes, and presenting the ability to bind HBV PreS1-derived peptides [16].

Despite many advantages, the HepaRG infection model remains restrictive. Notably, the long-term differentiation process of the cells may affect the reproducibility of the observed results. In addition, culture conditions are delicate, and the infection efficiency is low, which makes these cells an inadequate model for high-throughput studies [37]. Recently, Lucifora and colleagues proposed a new procedure allowing fast differentiation and efficient HDV-infection of HepaRG cells, opening the way to an easier use of this highly relevant model [42].

## 5. NTCP-Expressing Cell Lines 

### 5.1. Huh7 and HepG2-Derived Cell Lines 

Huh7 and HepG2 cells are two hepatoma-derived cell lines commonly used as a substitute model for hepatocytes, even if they only partially mimic hepatocyte functions [37]. In the context of HDV infection, these cells do not express NTCP and are therefore not susceptible to the virus [37]. Thus, these models do not allow a comprehensive understanding of the full viral cycle, including the early stages of viral entry and trafficking in hepatocytes [43,44]. By contrast, these cells can support complete HDV replication, as their co-transfection with plasmids encoding the HDV genome and HBV envelope proteins leads to the production of recombinant HDV particles able to infect susceptible cells [37]. 

Thanks to the confirmation of NTCP receptor importance in HBV and HDV infection of hepatocytes, Yan et al. demonstrated that NTCP-overexpressing Huh7 and HepG2 cells became susceptible to HDV infection [15], providing the first HDV-susceptible cancer-derived cell line suitable for high-throughput studies. 

Since their development, numerous studies have been conducted using NTCP-overexpressing cell lines to identify new factors involved in HDV infection and to characterize the different steps of the life cycle. Regarding viral entry, they allowed the characterization of the attachment factor GPC5 [24] and the involvement of IL6 in the regulation of NTCP expression [45]. Interestingly, these cells were also used to understand the involvement of NTCP in the entry process of another major hepatotropic virus, HCV [46]. Their ability to support high-throughput genetic screens was recently highlighted by the identification of 191 candidate factors for HDV infection from a druggable siRNA library. Among them were found the estrogen receptor alpha (ESR1) and the triple enzyme CAD involved in pyrimidine biosynthesis and playing a critical role in HDV replication [47]. NTCP-overexpressing cell lines also allowed researchers to highlight the interaction between HDV RNA and the innate immune sensor MDA5 that leads to a robust activation of the type I and III IFN pathway and the expression of several ISGs [48]. Elegantly pointing out an unexpected characteristic of this satellite virus, Giersch et al. took advantage of NTCP-overexpressing cells to demonstrate that HDV may persist by transmitting HDV RNA to dividing cells even in the absence of HBV coinfection. This strong persistence capacities of HDV could also explain why HDV clearance is difficult to achieve in HBV/HDV chronically infected patients [49]. 

In addition to the characterization of virus-related host factors or specific characteristics of the viral life cycle, these cell lines have been used to test antiviral agents, notably through targeting NTCP such as Myrcludex B, a lipopeptide derived from the HBV envelope protein, which has been shown to inhibit HBV and HDV entry in hepatocytes [50]. Additional small molecules, such as cyclosporin A [51,52], irbesartan [53,54], and vanitaracin A have demonstrated antiviral against HBV and HDV in NTCP-overexpressing cells [55]. 

Thus, these innovative cell culture systems are useful tools to improve our understanding of the HDV life cycle. Indeed, they have the advantage of infinite cells’ proliferation. However, a limit of these models concerns the genetic and metabolic differences compared to hepatocytes [37]. In addition, they require a high viral inoculum in contrast to infection under physiological conditions. Therefore, validation steps in alternative model systems are recommended, including PHH or suitable in vivo models such as human liver chimeric mice or transgenic mice expressing a chimeric version of NTCP [56,57]. 

### 5.2. Other NTCP-Overexpressing Hepatoma Cell Lines 

The key role of the NTCP cellular receptor in HBV and HDV entry has also allowed the development of other hepatoma cell lines overexpressing this factor, such as human cell line Li23-derived cells overexpressing NTCP (A8 cells subcloned from Li23 cells), whose gene expression profile is distinct from that of HepG2-NTCP or Huh7-NTCP cells. As the HBV susceptibility of A8 cells was far weaker than HepG2-NTCP cells, Ueda et al., successfully established a new cell line A8.15.78.10 exhibiting high HBV susceptibility comparable to HepG2-NTCP cells by repeated subcloning of A8 cells. The characterization of this new cell line demonstrated that the increase in HBV susceptibility was associated with an increase in the protein and glycosylation levels of NTCP, as well as to a reduction of STING expression [58]. 

In addition, by stably transducing HepG2 cells with genes encoding the NTCP-receptor and the HBV envelope proteins, Lempp et al. produce a cell line (HepNB2.7) that allows continuous secretion of infectious progeny HDV following primary infection. This cell line supports the complete HDV replication cycle and presents a convenient tool for antiviral drug evaluation [59]. The same group recently produced the Huh7-END cell line generated through stepwise stable integration of the cDNA of the HDV antigenome, the genes for the HBV envelope proteins and the HBV/HDV receptor NTCP. These cells can release HDV particles and are susceptible to de novo HDV entry. Thus, Huh7-END cells are a novel tool for the screening of antiviral drugs targeting HDV [60].

## 6. In Vitro Model Systems Based on Engineered Non-Hepatic Cell Lines

As an alternative strategy for the comprehensive characterization of the virus life cycle, recapitulating virus infection in non-hepatic cells may allow us to discriminate the key number of liver-specific factors required for virus infection. In this context, the expression of hNTCP in the non-hepatic human HeLa cells or in mouse-derived cells confers susceptibility to HDV while HBV is still restricted, suggesting that hNTCP, critical for viral entry, may be the only cell-specific factor limiting HDV infection [61].

Moreover, Yang et al. developed a non-hepatic cell culture model by exogenous expression of four host genes, which are the HBV/HDV entry receptor NTCP and the three nuclear hormone receptors HNF4α, RXRα and PPARα, in human non-hepatic 293T cells. 293T or HEK 293T is a human cell line derived from human embryonic kidney 293 cells (HEK293 cell line) that expresses a mutant version of the SV40 large T antigen, and that is commonly used in biology for protein expression or the production of recombinant retrovirus. The results obtained by Yang et al., indicated that this cell culture model supports HBV entry, transcription, and replication, as evidenced by the detection of HBV pgRNA, HBV cccDNA, HBsAG, HBeAg, HBcAg and HBV DNA. Interestingly, these 293T-NE-3NRs cells were successfully infected with HBV even at low GEq which mimics natural physiological conditions more accurately [62]. Contrary to HBV, HDV does not seem to require the expression of these hepatocyte-enriched transcription factors. Although not tested in the study, NTCP-expressing 293T cells should be susceptible to HDV infection.

## 7. In Vitro Model Systems Based on Induced Human Hepatocyte-Like Cells 

Recent advances in cell reprogramming techniques made possible the production of induced human hepatocyte-like cells (iHep). These cells are generated from induced pluripotent stem cells or by induced differentiation of human embryonic stem cells and exhibit at least some aspects of the innate immune response. They are expected to have more physiological characteristics of human hepatocytes than cell lines and can support the full life cycle of HBV, including the spread of the virus between cells due to the long-term maintenance of hepatic differentiation. iHep cells are virtually unlimited, appropriate for genetic manipulation, and recapitulate infected hepatocytes in humans [63,64,65]. Additional solid data including strategies to optimize or transdifferentiate pluripotent stem cells would be required to assess the ability of these systems to be deeply used for the study of HBV and HDV infection [66].

## 8. Conclusions and Perspectives 

Despite recent advances in the field, in vitro models for the study of HDV entry and infection remain imperfect and are individually limited by a series of constraints (Table 1). Indeed, primary hepatocytes are difficult to handle in culture and have a restricted capacity of infection by HDV. In the same way, HepaRG cells require specific culture conditions and a long differentiation process, thus limiting their availability for high-throughput studies [37]. In addition, the development of NTCP-overexpressing hepatoma cell lines has led to major discoveries in the understanding of the interactions between HDV and hepatocyte host factors. However, these cancer cells only possess a limited amount of human hepatocyte features, making them an interesting but limited model system in recapitulating the complex biology of hepatocytes [67]. Thus, other models for studying HDV and HBV infection have been developed to simulate the conditions of infection as closely as possible, such as systems based on induced human hepatocyte-like cells. 

In this context, systems based on organoid cultures have been recently developed as a substitute in vitro model to mimic tissues. This may help to join the gap between 2D cultures and in vivo mouse or human models [68]. Liver organoids have been produced for multiple species derived from induced embryonic stem cells, pluripotent stem cells, hepatoblasts, and adult tissue-derived cells. They recapitulate the complexity of the liver, maintain innate immune responses, and retain cell polarity, thus mimicking the natural entry of HBV and HDV. This system is an advancement in the models for generating fundamental knowledge of HBV and HDV biology and providing a promising platform toward screening potential new therapies and the development of customized hepatitis treatment [69]. However, these models require specific culture conditions [68] and solid data on their infection by HDV are still required at the time of this review.

To conclude, the recent advances in HDV and HBV biology provide a series of highly complementary models that can be used to cover of the aspects of the HDV life cycle through a broad battery of molecular approaches. The integration of different models coupled with validation in existing in vivo models may help to rapidly conduct a comprehensive overview of the molecular interactions between HDV and its specific host, for the development of new therapeutic strategies to tackle this worldwide health threat.

## Figures and Tables

**Table 1 viruses-13-01532-t001:** In vitro models used for the investigation of HBV and HDV entry and infection. The respective model system and their key benefits and limitations are shown.

Cell Model	Benefits	Limitations
PHH	Natural host of HBV/HDV Physiological model (Hepatic functions) Support complete virus life cycle Functional immune response	Low infection efficiency Donor variability Restrictive culture conditions Supply difficulties
PTH	More reproductible infections than PHH	Non-human cells and limited infection efficiency
HepaRG Cell Line	Exhibit a significant number of physiological liver functions Support infection	Limited infection efficacy Delicate culture conditions Long-term differentiation
Huh7-NTCP/ HepG2-NTCP	Easily available High reproducibility	Only partially mimic hepatocytes High viral inoculum
iHep Cells	Mimic natural hepatocytes Unlimited in supply Support full virus life-cycle	Complex differentiation conditions
**Other NTCP-Overexpressing Hepatoma Cell Lines:**		
Li23-derived cells	High susceptibility to HBV Different genetic background than HepG2 cells	Only partially mimic hepatocytes
Huh7-END	Stable HDV particle production Study of the full viral life cycle Screening of antiviral drugs	Only partially mimic hepatocytes
**Non-Hepatic Human Cell Lines:**		
HeLa-NTCP	Mimic more natural infection conditions Susceptibility to HDV but less to HBV	No hepatocyte functions
293T-NE-3NRs	High susceptibility to HBV Mimic more natural infection conditions	No hepatocyte functions Not yet tested with HDV

PHH: primary human hepatocytes; PTH: primary *Tupaia* hepatocytes; iHep cells: induced human hepatocyte-like cells.

## Data Availability

Not applicable.

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
