# Peer review of "Cell Culture Models for the Study of Hepatitis D Virus Entry and Infection"

_viruses, 2021, doi:10.3390/v13081532_

Round 1
Reviewer 1 Report
The manuscript by Heuschkel et al provides a review of the literature on different cell culture models for the study of hepatitis D virus infection. Hepatitis D virus infection, which is responsible for the most aggressive forms of viral hepatitis and for which an effective treatment is not yet available, remains an important health problem. Thus in this context, this concise and well documented review constitutes an important contribution to the field.
I have only a few minor comments which are listed below:
- table 1 should be completed with additional information, e.g. give specific information for each cell lines (293T-NE-3NR, HeLa-NTCP, Huh7-NTCP...); add Huh7-END, etc....
- the paragraph dealing with liver organoids should rather be moved to the conclusion/perspectives since no solid data are yet available on their infection by the hepatitis Delta virus.
I was able to note a few syntax errors throughout the text :
- line 90: correct "trough"
- line 119: replace "summary" by "summarize
- line 269: replace "NTCP-expression 293T cells" by "NTCP-expressing 293T cells
- line 273-275: words are missing in the sentence. Please rephrase
- Table 1: about non-hepatic human cell lines: correct "coniditions
Author Response
Point by point response to Reviewer 1
The manuscript by Heuschkel et al provides a review of the literature on different cell culture models for the study of hepatitis D virus infection. Hepatitis D virus infection, which is responsible for the most aggressive forms of viral hepatitis and for which an effective treatment is not yet available, remains an important health problem. Thus, in this context, this concise and well documented review constitutes an important contribution to the field.
I have only a few minor comments which are listed below:
- Table 1 should be completed with additional information, e.g. give specific information for each cell lines (293T-NE-3NR, HeLa-NTCP, Huh7-NTCP...); add Huh7-END, etc....
We agree with the reviewer that more information is required. The table 1 has been modified accordingly and specific information for each cell lines have been added.
- The paragraph dealing with liver organoids should rather be moved to the conclusion/perspectives since no solid data are yet available on their infection by the hepatitis Delta virus.
We agree with the reviewer that this paragraph should be modified. Thus, this part has been removed and the information about in vitro model systems based on liver organoids has been added in the perspectives as followed:
Line 296: “In this context, systems based on organoid cultures have been recently developed as a substitute in vitro model to mimic tissues. This may help to join the gap between 2D cultures and in vivo mouse or human models [69]. Liver organoids have been produced for multiple species derived from induced embryonic stem cells, pluripotent stem cells, hepatoblasts, and adult tissue-derived cells. They recapitulate the complexity of the liver, maintain innate immune responses, and retain cell polarity, thus mimicking the natural entry of HBV and HDV. This system is an advancement in the models for generating fundamental knowledge of HBV and HDV biology and providing a promising platform toward screening potential new therapies and the development of customized hepatitis treatment [70]. However, these models require specific culture conditions [69] and solid data on their infection by HDV are still required at the time of this review.”
- Minor points:
- line 90: correct "trough"
- line 119: replace "summary" by "summarize
- line 269: replace "NTCP-expression 293T cells" by "NTCP-expressing 293T cells
- line 273-275: words are missing in the sentence. Please rephrase
- Table 1: about non-hepatic human cell lines: correct "coniditions
We thank the reviewer for having pointed out these typos. They have been corrected accordingly.
Reviewer 2 Report
This is a useful, thorough and clear review of an important and topical area. It includes relevant recent models and references, so will be useful for others working in the field, particularly those entering HDV research from other viruses.
Although it is very well-written overall there are a few small grammatical errors scattered throughout that would benefit from thorough proof-reading by a native English speaker.
Author Response
Point by point response to Reviewer 2
This is a useful, thorough, and clear review of an important and topical area. It includes relevant recent models and references, so will be useful for others working in the field, particularly those entering HDV research from other viruses.
Although it is very well-written overall there are a few small grammatical errors scattered throughout that would benefit from thorough proof-reading by a native English speaker.
We thank the reviewer for his positive comment. The manuscript has been reviewed and grammatical errors have been corrected, as highlighted in red in the new version of the manuscript.

Reviewer 3 Report
Heuschkel et al. provide a review of cell culture model systems for the study of HDV. The review is well organized and touches on the main cell culture models currently available. Moderate editing of spelling and English required, but overall a nice piece of work.
My only comment is that the sections on iHeps and liver organoids could be expanded, since the authors hone in on these two platforms as the furthest advanced for the testing of HDV.
Author Response
Point by point response to Reviewer 3
Heuschkel et al. provide a review of cell culture model systems for the study of HDV. The review is well organized and touches on the main cell culture models currently available. Moderate editing of spelling and English required, but overall a nice piece of work.
My only comment is that the sections on iHeps and liver organoids could be expanded, since the authors hone in on these two platforms as the furthest advanced for the testing of HDV.
We thank the reviewer for this comment. At the time of the review, iHeps and liver organoids have not been widely studied for HDV infection, making difficult to expand these parts. To avoid confusion and insist on the novelty of these systems, the organoid section has been moved to the conclusion and perspectives section, which now reads as followed:
Line 296: “In this context, systems based on organoid cultures have been recently developed as a substitute in vitro model to mimic tissues. This may help to join the gap between 2D cultures and in vivo mouse or human models [69]. Liver organoids have been produced for multiple species derived from induced embryonic stem cells, pluripotent stem cells, hepatoblasts, and adult tissue-derived cells. They recapitulate the complexity of the liver, maintain innate immune responses, and retain cell polarity, thus mimicking the natural entry of HBV and HDV. This system is an advancement in the models for generating fundamental knowledge of HBV and HDV biology and providing a promising platform toward screening potential new therapies and the development of customized hepatitis treatment [70]. However, these models require specific culture conditions [69] and solid data on their infection by HDV are still required at the time of this review.”
Moreover, the following sentence was added to the “Induced Human Hepatocyte-like Cells” section:
Line 273: “Additional solid data including strategies to optimize or transdifferentiate pluripotent stem cells would be required to assess the ability of these systems to be deeply used for the study of HBV and HDV infection [67].”
Finally, as requested by the reviewers, we proof-read and edit the text, correcting typos and English grammatical errors, as highlighted in red in the text.
